# mTORC1 and SGLT2 Inhibitors—A Therapeutic Perspective for Diabetic Cardiomyopathy

**DOI:** 10.3390/ijms242015078

**Published:** 2023-10-11

**Authors:** Sumit Saha, Xianjun Fang, Christopher D. Green, Anindita Das

**Affiliations:** 1Department of Biochemistry and Molecular Biology, Virginia Commonwealth University, Richmond, VA 23298, USA; sahas3@vcu.edu (S.S.); xianjun.fang@vcuhealth.org (X.F.); christopher.d.green@vcuhealth.org (C.D.G.); 2Division of Cardiology, Pauley Heart Center, Department of Internal Medicine, Virginia Commonwealth University, Richmond, VA 23298, USA

**Keywords:** diabetes, diabetic cardiomyopathy, SGLT2i, mTORC1/2, AMPK

## Abstract

Diabetic cardiomyopathy is a critical diabetes-mediated co-morbidity characterized by cardiac dysfunction and heart failure, without predisposing hypertensive or atherosclerotic conditions. Metabolic insulin resistance, promoting hyperglycemia and hyperlipidemia, is the primary cause of diabetes-related disorders, but ambiguous tissue-specific insulin sensitivity has shed light on the importance of identifying a unified target paradigm for both the glycemic and non-glycemic context of type 2 diabetes (T2D). Several studies have indicated hyperactivation of the mammalian target of rapamycin (mTOR), specifically complex 1 (mTORC1), as a critical mediator of T2D pathophysiology by promoting insulin resistance, hyperlipidemia, inflammation, vasoconstriction, and stress. Moreover, mTORC1 inhibitors like rapamycin and their analogs have shown significant benefits in diabetes and related cardiac dysfunction. Recently, FDA-approved anti-hyperglycemic sodium–glucose co-transporter 2 inhibitors (SGLT2is) have gained therapeutic popularity for T2D and diabetic cardiomyopathy, even acknowledging the absence of SGLT2 channels in the heart. Recent studies have proposed SGLT2-independent drug mechanisms to ascertain their cardioprotective benefits by regulating sodium homeostasis and mimicking energy deprivation. In this review, we systematically discuss the role of mTORC1 as a unified, eminent target to treat T2D-mediated cardiac dysfunction and scrutinize whether SGLT2is can target mTORC1 signaling to benefit patients with diabetic cardiomyopathy. Further studies are warranted to establish the underlying cardioprotective mechanisms of SGLT2is under diabetic conditions, with selective inhibition of cardiac mTORC1 but the concomitant activation of mTORC2 (mTOR complex 2) signaling.

## 1. Diabetic Cardiomyopathy

Diabetes mellitus (DM) is a chronic metabolic disorder with a current prevalence of 573 million individuals worldwide, estimated to reach 643 million by 2030 and 783 million by 2045 [1]. Globally, around 18 million more men than women have diabetes [2] and the number of aged patients (>65 years) with DM is projected to reach ~276 million by 2045 as it is more prevalent in middle-aged and older adults [3]. DM ranks among the top 10 global mortality causes and is riddled with several health complications like cardiovascular dysfunction, leading to progressive heart failure. Cardiovascular diseases are the main cause of morbidity and account for two out of three overall deaths in diabetic patients [4], arising from conditions like hypertension, obesity, and dyslipidemia [5]. The concept of diabetes-related cardiomyopathy was first suggested in 1954 by Lundbæk [6]; later, in 1972, Rubler et al. reported the post-mortem findings of four diabetic patients who showed advanced symptoms of heart failure without any direct relation to congenital, valvular, or atherosclerotic heart conditions [7]. In 2013, to set a standard clinical perspective of diabetic cardiomyopathy, the American Heart Association [8] and the European Society of Cardiology [9] defined it as a pathophysiological condition of ventricular dysfunction in diabetic patients, without predisposing atherosclerosis and coronary artery disorders, or hypertensive, congenital, or valvular heart diseases.

DM can be broadly classified into type 1 diabetes mellitus (T1D) and type 2 diabetes mellitus (T2D). T1D, associated with autoimmune insulin deficiency, accounts for 5–10%, while T2D, characterized by insulin resistance, accounts for 90–95% of all diabetes cases [10]. Although cardiovascular complications are associated with both types of diabetes mellitus, the incidence of cardiac dysfunction in T1D and T2D patients has been reported to be 14.5% and 35%, respectively [11], making T2D-related cardiomyopathy the more prevalent pathophysiology.

The most common metabolic dysregulations in T2D, arising from insulin resistance [12], involve hyperglycemia and hyperlipidemia, which directly or indirectly provoke cardiac dysfunction in patients [13,14]. The heart shows reduced mitochondrial glucose oxidation due to insulin resistance along with excess fatty acid uptake, impaired mitochondrial fatty acid β-oxidation, and increased reactive oxygen species (ROS), resulting in lipotoxicity and cellular stress. These insults typically affect ventricular compliances with increased systemic pressure and manifest as impaired diastolic function with cardiac inflammation, abnormal calcium transport, and cardiac remodeling, which are linked to a slow progression into systolic dysfunction and ultimately heart failure [15,16,17,18] (Figure 1).

The systemic hyperglycemia, hyperlipidemia, oxidative stress, and inflammation associated with diabetes contribute to diabetic cardiomyopathy through several molecular pathways that provide significant therapeutic targets against diabetes-associated cardiac dysfunction. Some preclinical studies concerning diabetic cardiomyopathy have reported improved endothelial function, reduced cardiac fibrosis/hypertrophy, and reduced inflammation with metformin (via 5′ adenosine monophosphate-activated protein kinase, AMPK-dependent and AMPK-independent mechanisms) [19], tadalafil (phosphodiesterase 5, PDE5 inhibitor) [20], and MCC950 (nucleotide-binding oligomerization domain- leucine rich repeat- and pyrin domain-containing protein 3, NLRP3 inflammasome inhibitor) [21], respectively, while other preclinical studies with sulforaphane (nuclear factor erythroid 2-related factor 2, NRF2 activator) [22] and saxagliptin (dipeptidyl peptidase-4, DPP4 inhibitor) [23] showed cardioprotective benefits by ameliorating cardiac oxidative stress and lipotoxicity. Owing to these therapeutic targets, several clinical trials were conducted in the last decade with a focus on T2D patients (Table 1). However, these studies lacked an overall benefit curve in terms of both the glycemic and non-glycemic context of diabetic cardiomyopathy, emphasizing the critical need for identifying new molecular targets for the development of novel and more effective therapeutics.

## 2. Differential Role of mTORC1 and mTORC2 in Diabetic Cardiomyopathy

Although the central theme of T2D is highlighted as insulin resistance and hyperglycemia, several disorders in diabetes patients tend to be partially insulin responsive and therefore the glycemic aspect in the case of diabetes is just one part of a deranged metabolic network [34]. Obesity, hypertension, and cancers in diabetic patients have earlier been reported to reflect insulin-responsive pathology [35,36,37], thereby suggesting that insulin may promote, rather than benefit, non-glycemic disorders, particularly cardiovascular diseases [38], which account for the majority of mortality in patients with T2D [39]. In this context, it is crucial to recognize a target that links the glycemic and non-glycemic aspects of diabetes to effectively treat diabetic cardiomyopathy.

The mammalian target of rapamycin (mTOR), a member of the phosphatidylinositol 3-kinase (PI3K)-related protein kinase superfamily, is crucial for insulin and insulin-like growth factor 1 (IGF-1) signaling and plays an important role in cell growth, proliferation, autophagy, apoptosis, inflammation, and metabolism. There are two distinct mTOR complexes, termed mTORC1 and mTORC2, with a common core catalytic subunit that harbors specific mTOR-interacting units that designate specific cellular functions to these complexes. mTORC1 has three core components: regulatory subunit Raptor (regulatory-associated protein of mTOR), catalytic subunit mTOR, and mLST8/GβL (mammalian lethal with SEC13 protein 8/G protein beta subunit-like). Raptor helps in substrate recruitment to the complex and ensures proper subcellular localization. mLST8 associates with the catalytic domain and stabilizes the kinase activation loop. Besides these components, mTORC1 has two inhibitory subunits PRAS40 (proline-rich Akt substrate of 40 kDa) and DEPTOR (disheveled EGL-10 and pleckstrin (DEP)-domain containing mTOR-interacting protein). mTORC2 consists of mLST8 and the catalytic mTOR subunit, but Raptor is replaced by an analogous subunit Rictor (rapamycin-insensitive companion of mTOR). mTORC2 also contains the inhibitory DEPTOR subunit along with other regulatory subunits mSin1 and Protor1/2. mTORC1 is extensively involved in protein synthesis, nucleotide synthesis, lipid synthesis, autophagy, and mitochondrial biogenesis, while mTORC2 is associated with cell survival, apoptosis, cytoskeletal organization, and glucose metabolism (Figure 2). The mTOR complexes and their upstream/downstream signaling processes have been extensively discussed in several reviews with respect to various pathophysiologies [40,41,42,43].

In the glycemic context, insulin resistance, driven by mTORC1 hyperactivation-mediated deregulation of the insulin receptor (IR)-PI3K/Akt substrate (IRS) signaling axis, leads to elevated blood glucose levels (hyperglycemia). Hyperactivation of mTORC1 and its downstream S6K1 kinase (Ribosomal S6 kinase) phosphorylates IRS-1 at Ser307 and Ser636/639 to initiate IRS-1 degradation [44] and PI3K/Akt signaling suppression, thereby causing insulin resistance via the mTORC1/S6K1 negative feedback loop. mTORC1 also regulates insulin signaling via Grb10 (growth factor receptor-bound protein 10), which inhibits threonine phosphorylation of insulin/IGF receptors and blocks PI3K/Akt signaling [45], thereby disrupting the IRS axis leading to insulin resistance and increased hyperglycemia [46]. 

In the non-glycemic context, closely associated with diabetic cardiomyopathy and endothelial disorders, mTORC1 has been prompted as a key player. The mTORC1 pathway has been linked to cardiac hypertrophy and hypertension. An upregulated mTORC1/S6K1 activity contributes to deregulated insulin-stimulated vasodilation by suppressing eNOS (endothelial nitric oxide synthase), resulting in vasoconstriction and hypertension. Several reports of diabetic cardiomyopathy and heart failure also demonstrate upregulated mTORC1 activity, whereas studies concerning mTORC1 inhibitors (rapamycin and PRAS40) [47,48] and induced cardiac autophagy (inhibited by mTORC1) show beneficial effects in diabetic cardiac dysfunction, implying a pathogenic role of mTORC1 hyperactivation [49]. In T2D patients, ischemia and cardiomyopathy go hand-in-hand, leading to heart failure. The condition results in a fibrotic phenotype of the heart which initially faces ejection fraction-preserved diastolic dysfunction and develops systolic dysfunction in the later stages [50]. mTORC1 regulates ischemic injury conditions by preserving energy homeostasis, and, as per literature, Rheb (Ras homolog enriched in brain) inhibition, which subsequently inhibits mTORC1 and protects cardiomyocytes by activating autophagy during energy deprivation and ischemia [51]. Studies have indicated that AMPK inhibition in glucose-deprived and ischemia conditions, resulting in mTORC1 hyperactivation, lead to worsening of cardiac dysfunction and cardiomyocyte death [52], while AMPK activation attenuates pressure overload or diabetes-related cardiac remodeling [53,54]. Moreover, AMPK activators like resveratrol, berberine, and metformin have also been associated with cardiovascular benefits in T2D. Resveratrol has been linked to upregulated adiponectin levels to prevent myocardial ischemia injury in diabetic mice [55], whereas berberine studies in diabetic cardiomyopathic rats have showed attenuated hypertrophy via activated AMPK and reduced GSK3β (glycogen synthase kinase 3 beta) [56]. A recent study by Yang F. et al., in 2019, further emphasized the role of AMPK activators in diabetic cardiomyopathy benefit and reported metformin-mediated NLRP3 inflammasome inhibition via AMPK/mTOR pathway [57]. These reports cumulatively suggest a critical role of mTORC1 downregulation in diabetic cardiomyopathy benefits. Figure 3 depicts mTOR signaling and its hyperactivation in cardiomyocytes that contributes to hyperlipidemia, inflammation, and stress.

Ionic imbalance, including a state of calcium overload as well as increased intracellular sodium, is a key player in the development of cardiac dysfunction and characteristics of diabetic cardiomyopathy [58,59,60]. mTOR signaling is involved in diverse biological pathways by regulating ionic homeostasis, specifically by regulating the activity and expression of various Ca^2+^ channels [61,62,63]. Intertwined links between sarcoplasmic reticulum calcium homeostasis and mTORC1 signaling are critical for physiological and pathological cardiac hypertrophy [64,65]. Inhibition of mTORC1 with rapamycin induces Ca^2+^ release from lysosomes through the activation of two-pore segment channel 2, TPC2 [66]. The downregulation of the inward rectifier potassium (I_K1_) channel with intracellular Ca^2+^ overload is a hallmark in cardiac hypertrophy, interstitial fibrosis, and electrical remodeling and failure [67]. Specifically, mTORC1 regulates the lysoNa_ATP_, which determines the sensitivity of endolysosome’s resting membrane potential to Na^+^ and cytosolic ATP as well as controls lysosomal pH stability and whole-body amino acid homeostasis [68,69]. Selective I_K1_ agonist attenuates cardiac remodeling by promoting autophagy via negatively regulating calcium-activated CaMKII and mTOR signaling [69]. The activation of I_K1_ channel protects the heart against myocardial ischemia-induced cardiac dysfunction by inhibiting mTOR-p70S6 signaling pathway [68]. mTOR also acutely controls endosomal and lysosomal functions through the endolysosomal ATP-sensitive Na^+^ channel (lysoNa_ATP_) in response to changes under different nutrition status and metabolic conditions [62]. mTORC1 regulates the endolysosomal ATP-sensitive Na^+^ channel (lysoNa_ATP_), which determines the sensitivity of endolysosome’s resting membrane potential to Na^+^ and cytosolic ATP as well as controls lysosomal pH stability and whole-body amino acid homeostasis. Under nutrient deprivation, mTORC1 interacts with lysosomal TPC2 and regulates authophagy [62,70]. 

From a therapeutic perspective, direct and indirect mTORC1 inhibitors have been broadly used for treating T2D and co-existing diabetic cardiac conditions. Metformin is a widely used, FDA-approved anti-diabetes drug that regulates mTORC1 via mitochondrial complex I-mediated AMPK activation [71] or AMPK-independent Rag GTPase inhibition [72]. Recent preclinical reports on metformin have shown attenuated myocardial hypertrophy [73] and reduced inflammation [57] in diabetic animal models but earlier clinical trials with metformin, as an overall cardioprotective drug in diabetic patients, are inconspicuous [19]. A 2014 study by Mirko Volkers et al. on PRAS40, a direct mTORC1 inhibitor, reported diabetic cardiomyopathy prevention besides improved hepatic insulin sensitivity in a diabetic mouse model [74] but further PRAS40 studies in diabetic cardiac dysfunction are yet to emerge. 

Rapamycin, another direct mTORC1 inhibitor, has been closely linked to improved T2D and diabetic cardiac dysfunction [75,76,77], but multiple studies have reported controversial effects of chronic treatment with rapamycin [78,79]. Studies with analogs of rapamycin or rapalogs, like everolimus, have shown beneficial effects and an improvement in glucose metabolism in diabetes by disrupting the mTORC1/S6K1 feedback loop [80], but similar to rapamycin, chronic treatment with rapalogs has shown detrimental effects in T2D patients [78]. These poor outcomes of chronic rapamycin treatment might involve the inhibition of mTORC2 activity [81]. However, chronic treatment with a sub-clinical dose (0.25 mg/kg/day) of rapamycin or nano-formulated micelles of rapamycin, rapatar, has ameliorated the metabolic status of diabetic mice, with an improvement in cardiac function by preferentially inhibiting mTORC1 [75,82]. Our studies on rapamycin also reported mTORC2 activation in diabetic mice and rabbits, which might be associated with improved cardiac function and reduced myocardial infarction following ischemia-reperfusion injury [83,84]. A 2014 study using an ischemia-reperfusion injury mouse model suggested that miR-144 improves cardioprotection via suppressing mTORC1 and simultaneously activating mTORC2 [85], while another study implicated the role of mTORC2 in preserving cardiac function in pressure-overload hypertrophy [86], therefore highlighting mTORC1 inhibition, alongside mTORC2 activation, as crucial mechanisms to consider in diabetic cardiomyopathy treatment.

## 3. Sodium and Glucose Co-Transporter Inhibitors (SGLT2is)—Do They Regulate mTORC1 in Diabetic Cardiomyopathy?

Despite the boom in preclinical and clinical studies on diabetic cardiomyopathy, its pathogenesis and unified paradigm remain unclear for devising specific therapeutic strategies. As a result, a plethora of research is still based on figuring out the most effective therapeutic approach to treating diabetes, diabetic cardiomyopathy, and related heart failure. To fill this critical gap in therapeutics, mTORC1 might be a key target for diabetic therapeutics, which also addresses other co-morbidities like cardiomyopathy in a glycemic and non-glycemic context. In T2D, metformin is usually the first-line standard treatment [87] and though a few studies have shown metformin to exert cardioprotective effects via mTORC1 regulation [57], these reports are very limited despite a long history of metformin use [88]. Recently, sodium–glucose co-transporter 2 inhibitors (SGLT2i), a new class of FDA-approved [89] anti-diabetic drugs, have shown a promising risk reduction of cardiomyopathy and other cardiovascular diseases in patients with T2D [90,91], owing to the overexpression of SGLTs in diabetes mellitus [92].

Glucose homeostasis, a crucial diabetes parameter associated with mTORC1 [93], is actively regulated by transporters like GLUTs and SGLTs that mediate D-glucose transport. While SLC2 genes encode for facilitated glucose diffusion transporters GLUT, sodium glucose co-transporters SGLT1-5 are encoded by SLC5. SGLT transports glucose into cells via Na^+^/K^+^-ATPase pump gradient and two major SGLT isoforms are SGLT1 and SGLT2. SGLT1 is expressed in the small intestine, kidneys, brain, and heart, while SGLT2 is expressed in kidney and pancreatic β cells. SGLT1 primarily acts as rate limiting intestinal glucose absorption whereas SGLT2 manages bulk glucose reabsorption in the kidneys [94]. In the kidneys, SGLT2 in the S1/S2 segment of the convoluted proximal tubule in kidney nephrons reabsorbs 90% of the glomerular filtrate glucose, aided by a positive sodium gradient [95], while SGLT1 reabsorbs the remaining 10% in the S3 segment of the proximal tubule [96]. The reabsorbed glucose in the tubular epithelial cells is flushed back into circulation through GLUT2 and this whole unidirectional transport of glucose is coupled to and regulated by the Na^+^K^+^ ATPase pump on the basolateral side of the cells [97]. SGLT2is primarily work as anti-hyperglycemic agents by blocking the SGLT2 channel and preventing glucose reabsorption, thus preventing hyperglycemia in diabetes mellitus. Preclinical studies with phlorizin, the first SGLT2 inhibitor, in the 19th century, improved insulin sensitivity in diabetic rat models but did not have any scope for oral bioavailability and showed adverse gastrointestinal concerns [98]. In 1990, T-1095, a phlorizin derivative, was developed as the first orally available SGLT2 inhibitor but was discontinued after phase II clinical trials owing to its non-selective nature and safety concerns [98]. In current therapeutic use, the common FDA-approved SGLT2is are empagliflozin, dapagliflozin, and canagliflozin, which have demonstrated cardiorenal benefits in diabetic patients [99,100]. 

Over the years, besides significant glucose-lowering efficacy, SGLT2is have also shown remarkable cardiovascular benefits in renal-impaired patients with lower glomerular filtration rates, indicating a major role of SGLT2is in promoting diabetic cardiac dysfunction benefits [101]. Table 2 provides a summary of SGLT2i clinical trials (phase III) in assessing cardiac dysfunction and heart failure with diabetes, while current trials are also focusing on SGLT2is in heart failure mediated by other metabolic insults like obesity [95,102]. A large multinational observational cohort study on T2D patients, CVD-REAL, associated the early initiation of SGLT2is with a lower risk of heart failure, myocardial infarction, and stroke, compared to other glucose-lowering drugs [100,103,104]. The EMPA-REG trial [100,105] in T2D patients with established cardiovascular risks showed a reduction in cardiovascular-related deaths with empagliflozin, while the DAPA-HF trial [106] in diabetic/non-diabetic patients with heart failure reported that dapagliflozin reduces the worsening of cardiovascular risk and heart failure by 26%, regardless of T2D status. These clinical trials suggested a moderate/no risk of genital infections with SGLT2i treatment but, interestingly, amputation risks were closely associated with SGLT2is like canagliflozin and ertugliflozin, which have a higher degree of non-selectivity towards SGLT2. Compared to other anti-diabetic drugs like DPP4i, safety analyses of SGLT2i treatment indicate a higher risk of genital infections, urinary tract infections, hypertension, and diabetic ketoacidosis, and a lower risk of acute kidney injury and decreased bone mineral density [107]. Other clinical trials concerning diabetic cardiomyopathy like the EMPEROR-Reduced trial [108] of SGLT2-selective empagliflozin also show reduced cardiovascular risks and heart failure in diabetic and non-diabetic patients with uncomplicated adverse genital infections, thus strongly suggesting that SGLT2is work in both glycemic and non-glycemic contexts. Therefore, deciphering the mechanism of SGLT2is and whether it concerns mTORC1 regulation in terms of cardiac dysfunction is crucial to understand its therapeutic role in diabetic cardiomyopathy.

Although SGLT2is have shown interesting benefits in terms of cardiac function with or without T2D, the expression of SGLT2 channels is negligible in the heart, further highlighting SGLT2-independent functions of these inhibitors in diabetic cardiomyopathy and cardiac dysfunction [134,135]. The presence of SGLT1 in the heart is well established and is reportedly overexpressed in T2DM patients [136]. Along with glucose transporters (GLUT), SGLT1 is involved in cardiomyocyte glucose uptake, and in ischemic conditions, SGLT1 is reported to supplement the ATP reserve by increasing glucose utilization, thus playing a role in myocardial energy metabolism [137,138]. Several clinical trials with SGLT2is have emphasized cardiac benefits, but most of these drugs show a variable affinity for SGLT1. To provide some evidence on specificity, Kondo et al. reported that non-selective SGLT2is, like canagliflozin, can mediate anti-inflammatory and anti-apoptotic effects, which are related to SGLT1-binding-mediated improved NOS coupling in cardiomyocytes, thus indicating SGLT1 inhibition by SGLT2is [138,139]. Studies on dapagliflozin acting as a SGLT1/2 dual inhibitor, also suggest the involvement of myocardial SGLT1 in mediating SGLT2i effects [140]. Some researchers therefore emphasize that SGLT1/2 dual inhibitors provide greater cardiac benefit and heart failure prevention compared to selective SGLT2is [141], but the role of SGLT1 inhibition in terms of cardiovascular benefits has a fair share of contradictory reports. While some studies reported reduced heart failure incidence [142] and enhanced AMPK [139] with SGLT1 inhibition, others showed that AMPK activation resulted in increased SGLT1 expression, which can lead to hypertrophy and ischemia [143]. To bypass these contradictions, trials to treat diabetic cardiomyopathy or cardiac dysfunction (without diabetes) use empagliflozin, which is more SGLT2-selective as compared to canagliflozin and dapagliflozin, thus limiting the chances of SGLT1 channel cross-targeting [144,145,146]. For such selective SGLT2i scenarios, off-target effects on GLUT receptors might be involved. While some researchers have hypothesized that the SGLT2i-GLUT binding inhibits glucose uptake in cardiac tissue in an SGLT1/2-independent manner [147] and empagliflozin has been proposed to dock on GLUT1 and GLUT4 [148], confirmed research is yet to be documented in terms of cardiac pathophysiology.

Amidst some theories to explain the role of SGLT2is in cardiac function that involves the renin–angiotensin–aldosterone system (RAAS) and diuretic pathways [149], a prominent hypothesis to support the role of SGLT2is in the heart concerns the involvement of sodium ion homeostasis [150]. Cardiac Na^+^ and Ca^2+^ homeostasis plays a major role in maintaining heart physiology, rhythm, and contraction [151]. Increased intracellular sodium (Na_i_^+^) due to hyperactive sodium hydrogen exchanger (NHE) is well known in cardiac dysfunction pathologies and leads to elevated Ca^2+^ efflux from the mitochondria, resulting in oxidative stress and deteriorated cellular function [152,153]. Besides variable degrees of cross-reactivity with SGLT1 [154], several reports on SGLT2is have demonstrated non-SGLT2 cardiac-based off-target effects in reducing ventricular myocyte systolic Ca^2+^ and lowering myocardial cytoplasmic Na^+^/Ca^2+^ via NHE regulation [155]. A reduction in myocardial Na_i_^+^ via the inhibition of Na^+^/H^+^ or Na^+^/Ca^2+^ exchangers has been reported to improve cardiac hypertrophy and heart failure [156,157]. Preclinical studies have shown that SGLT2is can directly bind and reduce NHE activity in cardiomyocytes, leading to decreased Na_i_^+^ and restored Ca^2+^ homeostasis, resulting in improved cardiac function and antioxidative capacity of cardiomyocytes [158,159], but the observations are not consistent [148,160]. Baartscheer et al. showed direct effects of empagliflozin on Na^+^ and Ca^2+^ alteration, independent of SGLT2 binding in isolated ventricular cardiomyocytes, thus indicating a similar role of empagliflozin to that of NHE inhibitors [158]. SGLT2is like empagliflozin and dapagliflozin have also been implicated in enhanced sarcoplasmic endoplasmic reticulum Ca^2+^-ATPase (SERCA2a) activity, which improves cardiac contractility via sarcoplasmic reticulum Ca^2+^ reuptake [161,162]. Under normal physiological conditions, the Na^+^/H^+^ exchanger pumps Na^+^ inside the cell and H^+^ outside the cell to maintain ionic homeostasis but in case of prolonged NHE activation during T2D, excess Na_i_^+^ increases sodium–calcium exchanger (NCX)-mediated intracellular Ca^2i+^, leading to oxidative stress and an acidic intracellular environment [163,164]. Therefore, by reducing NHE hyperactivity, SGLT2is can promote an alkaline intracellular environment. A recent bioRxiv preprint study by Kazyken and colleagues reported that alkaline intracellular pH can activate the AMPK/mTORC2 pathway and inhibit mTORC1 activity [165]. This links the role of SGLT2is to AMPK activation and subsequent mTORC2 activity via NHE-mediated pH homeostasis and Ca_i_^2+^ modulation [64]. 

From another perspective, SGLT2 inhibitors have been shown in some studies to promote ketogenesis, lipid oxidation, and erythrocytosis, which can reflect a fasting-like transcriptional paradigm by mimicking nutrient deprivation and hypoxia [166] but can also be responsible for the moderate adverse effects in SGLT2i clinical trials (Table 2). Although systemic glucose lowering or ketogenesis by SGLT2is in vivo can modulate a starvation and nutrient deprivation environment, a possible GLUT inhibition by selective SGLT2is might be responsible for glucose deprivation in isolated cardiomyocytes. A depletion in the glucose environment caused by SGLT2is can reduce the ATP/ADP ratio, leading to increased AMP that stimulates the phosphorylation of AMPK and subsequently, phosphorylates GAPDH to activate SIRT1 [167]. AMPK (nutrient sensor) and SIRT1 (redox rheostat) activation help cardiomyocytes to adapt in response to SGLT2i-mediated nutrient-deprived conditions. AMPK and SIRT1 activation have been reported to negatively regulate mTORC1 in a Tsc1/2-dependent manner [49,168], thus indicating that SGLT2is indirectly inactivate mTORC1 by mimicking a nutrition-deprivation setting. Furthermore, a recent study of SGLT2is in obesity-related diabetic cardiomyopathy reported sestrin2-mediated AMPK activation/mTORC1 inactivation in cardiomyocytes upon empagliflozin treatment [169]. Some studies have reported energy deprivation as a cause for sestrin2 activation [170], which can be correlated with AMPK/mTORC2 activation besides inhibiting mTORC1 and promoting autophagy [171]. This adds another revelation to the role of SGLT2is in attenuating diabetic cardiomyopathy conditions by promoting an energy-deprived environment. A recent study by Zhang et al. in 2023 suggested that empagliflozin significantly reduced diabetic cardiomyopathy by promoting branched-chain amino acid catabolism, inhibiting mTORC1/p-ULK1, and reactivating autophagy [172], while another study by Feng et al. reported that dapagliflozin prevented cardiac dysfunction in diabetic rats by restoring autophagy by repressing mTORC1 and activating AMPK [173]. Figure 4 shows a summary of SGLT2 inhibitor mechanisms to target renal glucose absorption (canonical) and cardiac mTORC1 signaling (non-canonical) in diabetic cardiomyopathy.

SGLT2is have been broadly assertive in terms of cardiac benefits via potential mechanisms of anti-inflammation, oxidative stress reduction, and apoptosis prevention [174]. Although the cardioprotective effects of SGLT2is were previously regarded as glucose-lowering systemic effects, current research points out several direct cardiac effects of SGLT2is and mTORC1 might be the missing link. While Leet et al. reported that dapagliflozin reduced inflammatory cytokines IL-6/IL-1β and superoxide levels in a myocardial infarction model [175], Shi X et al. showed attenuation of pro-inflammatory cyclooxygenase-2 and IL-1β in a heart failure model [176]. Taking a step deeper into the mechanism, Ye Y et al. showed that dapagliflozin reduced diabetic-induced activation of cardiac nucleotide-binding oligomerization domain-like receptor 3 (NLRP3) inflammasome and the subsequent stimulation of pro-inflammatory cytokine production, which are associated with T2DM cardiac inflammation [177]. The researchers also found that dapagliflozin reduced apoptosis speck-like protein containing a caspase recruitment domain (ASC) and IL-1β in cardiofibroblasts in vitro, indicating that these effects are not SGLT2-related or glycemic. Besides inflammation, oxidative stress is a major player in diabetic cardiomyopathy and cardiac hypertrophy [178]. High doses of empagliflozin have been reported in some studies to reduce cardiac superoxide levels, advanced glycation end products (AGE), and AGE receptors (RAGE) in diabetic female rodent models [179], while dapaglifozin in myocardial infarction models has been implicated as an antioxidant modulator through direct reactive oxygen and nitrogen species (RONS)-dependent STAT3 signaling, independent of SGLT2-binding or glucose-lowering anti-diabetic effects [175]. Endoplasmic reticulum stress pathway (ERS)-mediated cardiac apoptosis via ROS is also a prominent pathological condition in diabetic cardiomyopathy and studies on empagliflozin have reported decreased ERS-associated caspase-12 [180]. However, some studies on different SGLT2i dosages show no anti-apoptotic benefit [162], making the evidence inconclusive. Although more studies are required to conclude the direct role of SGLT2is in NLRP3, ROS, and ERS-associated signaling, SGLT2i-mediated mTORC1 regulation can be hypothesized as the medium. Several reports indicate that mTORC1 activation induces NLRP3 inflammasome, while rapamycin and AMPK activation can inhibit mTOR/NLRP3 [57,181]. Moreover, literature reports also establish AMPK-mediated STAT3 inhibition via attenuation of JAK signaling and activation of redox-regulating NRF2 [182]. Therefore, aside from the systemic glucose-lowering advantage of SGLT2i cardioprotection, mTORC1 regulation by selective SGLT2is can also explain the direct mechanism of documented SGLT2i action in cardiac tissue to prevent cardiac dysfunction in diabetic cardiomyopathy patients.

## 4. Future Directions

With the advent of SGLT2is as potential anti-diabetes drugs with independent cardioprotective effects, several preclinical comparative studies with existing FDA-approved drugs, like metformin, sulfonylurea, DPP4i-inhibitors, and GLP-1 agonists, have come to the forefront. Sulfonylureas are the oldest form of anti-diabetic drug which stimulate insulin from pancreatic beta cells [183], while metformin is regarded as the first line of diabetic drugs, which reduces glucose production in the liver and enhances insulin sensitivity [184]. DPP4i and GLP-1 agonists both work by stimulating insulin secretion after an oral glucose load via incretin effect [185,186]. An observational multidatabase cohort study reported reduced myocardial infarction, stroke, and heart failure (MACEs—major adverse cardiovascular events) with SGLT2is as compared to DPP4is [187], whereas another database study concerning SGLT2is vs. metformin reported a trend of decreased heart failure hospitalizations and mortality events with SGLT2is in T2D patients [188]. Other cohort studies in terms of combination therapy showed SGLT2i–metformin to have a reduced all-cause mortality risk compared to SGLT2i monotherapy or sulfonylurea–metformin [189]. Recently, several observational comparative studies are focusing on SGLT2i and GLP-1 agonists as they are the first anti-diabetic drugs to demonstrate definite direct cardiac benefits with a reduction in glycated hemoglobin level in T2DM [190]. In a meta-analysis of several cardiovascular outcome trials regarding SGLT2i and GLP-1 agonists, Zelniker et al. found that MACE reduction was restricted to SGLT2i-administered patients with established atherosclerosis [191], while Wright et al. demonstrated that both SGLT2i monotherapy and SGLT2–GLP-1 agonist combination therapy may have a beneficial primary MACE risk reduction [192]. Emphasizing mTORC1 inhibition/mTORC2 activation as our proposed key to cardiovascular improvements in diabetic cardiomyopathy, several reports of GLP-1 agonists show mTORC2 activation [193] besides SGLT2i-mediated mTORC1 inhibition, making their combination therapy ideal, but there are limitations of increased hypoglycemia risk [194]. To date, there are no randomized controlled trials that compare SGLT2is with GLP-1 agonists head-to-head in diabetic cardiomyopathy and hence the current inconsistent meta-analysis-based interpretations have their limitations, thus demanding further preclinical and clinical studies in the future.

Besides diving into a deeper cardiovascular understanding of SGLT2is, there is a dire need to focus on the adverse effects of SGLT2i treatment in diabetic patients. The association of SGLT2i treatment with genital infections has been reported in several trials like EMPA-REG [109], DECLARE-TIMI [118], and EMPEROR-Reduced [130], which might be due to higher glucose concentrations in the urine, which promote bacterial growth [195,196], but the statistical data are inconclusive. Currently, a one-year observational study is recruiting female T2DM patients taking empagliflozin or dapagliflozin to correlate SGLT2is with urinary tract infections [197], but further studies are required to document statistical significance and mitigate any severe adverse effects of SGLT2i monotherapy or combination therapy.

With several completed and ongoing clinical trials, SGLT2 inhibitors are rapidly emerging as the miracle anti-diabetic drug. Along with beneficial renal outcomes and reduced kidney insults in T2DM patients, SGLT2is have also galvanized their position as a potential treatment candidate for diabetic cardiomyopathy and other cardiac disorders owing to their direct SGLT2-independent cardiac effects, thus encouraging further applications of SGLT2is in other metabolic co-morbidities like non-alcoholic liver steatohepatitis (NASH), with or without diabetes/obesity. A placebo-controlled interventional phase II study (LEGEND) by Inventiva Pharma is currently recruiting participants to compare the effects of lanifibranor (a pan-peroxisome proliferator-activated receptor agonist) monotherapy and lanifibronor–empagliflozin combination therapy in patients with NASH and T2DM [198], whereas another phase IV interventional clinical trial is underway to assess the effect of empagliflozin on fatty liver in non-diabetic patients [199]. With several other preclinical and clinical trials [200,201,202] lined up to assess the potential of SGLT2 inhibitors in various metabolic pathophysiologies, our review presents a crucial molecular perspective of SGLT2is’ mechanism of action mediated by mTOR complexes. A better understanding of how off-target SGLT2i effects can regulate mTOR complexes might be the stairway to repurposing these miracle drugs in the near future.

## 5. Conclusions

Cardiovascular diseases like diabetic cardiomyopathy are critical co-manifestations in patients with diabetes mellitus and several medical approaches are currently targeting diabetes with a significant consideration for treatments that also improve cardiac dysfunction and cardiomyopathy. A thorough analysis of T2D and its co-morbidities, in both glycemic and non-glycemic contexts, translates into mTORC1 being a unified therapeutic target. Besides the standard metformin treatment for diabetes, SGLT2 inhibitors have come up recently as potential anti-diabetic drugs that show very promising cardiovascular protection. Although the mechanisms of the cardiac effects of SGLT2is are still being explored, the existing hypotheses consistently point to mTORC1 regulation via ionic dyshomeostasis/stress and mimics of nutrient deprivation. While the role of SGLT2is in downregulating mTORC1 is very critical for cardioprotective effects in diabetic patients, owing to the evidence of improved cardiac function by mTORC1-inhibitor rapamycin, all roads do not lead to Rome; several studies have indicated that a fine balance between mTORC2 activation and mTORC1 inhibition is optimal for cardiac benefits in patients with/without diabetes. An in-depth understanding of off-target SGLT2i effects might open up novel treatment regimes with SGLT2is in several other cardiac, renal, pancreatic, cerebral, and hepatic pathologies as a single drug or combination therapy. With further clinical progress in exploring the mechanisms of how SGLT2is regulate cell signaling, drug modifications might also help in bypassing the adverse effects of genital infections, thus ameliorating the standard of patient care. Therefore, further research is pivotal to understand novel mechanisms of SGLT2is in the non-glycemic SGLT2-independent context besides shedding light on the role of SGLT2is in regulating the mTORC1/mTORC2 switch for a holistic approach towards diabetic cardiomyopathy and related metabolic disorders.

## Figures and Tables

**Figure 1 ijms-24-15078-f001:**
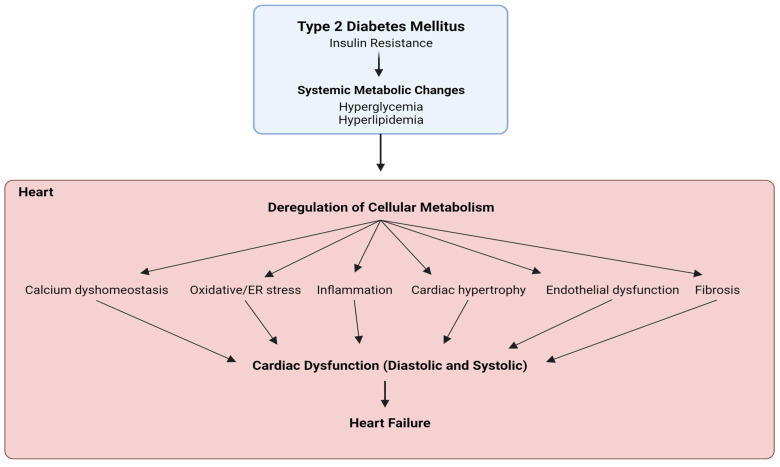
**Schematic representation of diabetic cardiomyopathy pathophysiology.** Insulin resistance in type 2 diabetes mellitus mediates systemic hyperglycemia and hyperlipidemia. These conditions induce metabolic changes in the heart and endothelial system, leading to mitochondrial dysfunction causing calcium imbalance and oxidative stress. As a result, other insults like inflammation, hypertrophy, and fibrosis arise as interdependent factors and culminate into cardiac dysfunction and progressive heart failure.

**Figure 2 ijms-24-15078-f002:**
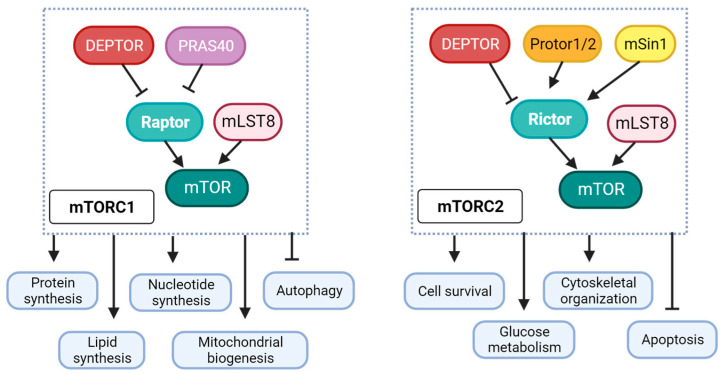
**mTORC1 and mTORC2 complexes and downstream cellular functions.** mTORC1 is composed of a core complex of mTOR, Raptor, and mLST8, which is inhibited by DEPTOR and PRAS40. mTORC2 comprises a core of mTOR, Rictor, and mLST8 that is inhibited by DEPTOR and regulated by Protor1/2 and mSin1. mTORC1 versus mTORC2 activation affects diverse cellular functions.

**Figure 3 ijms-24-15078-f003:**
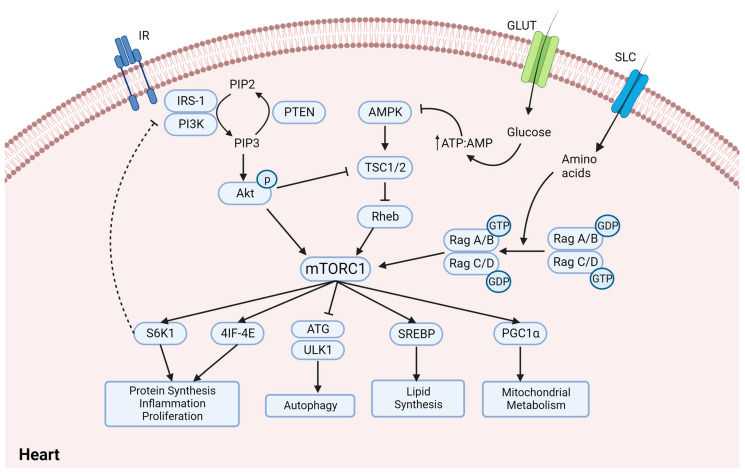
**The cardiac mTORC1 signaling network.** In normal conditions, IR-mediated PI3K/Akt activation leads to activated mTORC1, which inhibits autophagy via ULK1 and promotes protein synthesis of inflammatory and proliferative markers via S6K1 and eIF4E. In diabetic cardiomyopathy, mTORC1 is hyperactivated due to AMPK inhibition by hyperglycemia-mediated high APT:AMP ratio and mTORC1 hyperactivation induces a negative feedback loop to inhibit Akt, resulting in insulin resistance. Besides insulin resistance, mTORC1 hyperactivation also leads to dysregulated lipid synthesis and mitochondrial biogenesis, resulting in ROS upregulation and cardiac dysfunction. eIF4E—eukaryotic initiation factor 4E; IR—insulin receptor; IRS1—insulin receptor substrate 1; GLUT—glucose transporter; SLC—solute carrier group of membrane transporters; Rag—Ras-related GTPase; Rheb—Ras homolog enriched in brain; ULK—Unc-51-like kinase; ATG—autophagy-related protein.

**Figure 4 ijms-24-15078-f004:**
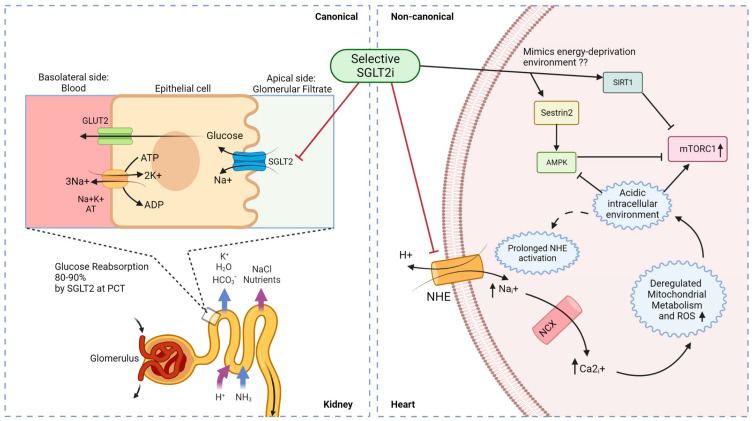
**Canonical and non-canonical (cardiac) mechanism for SGLT2i function in diabetic cardiomyopathy**. Luminal glucose in the renal glomerular filtrate is reabsorbed (80–90%) into the proximal tubular epithelial cells via SGLT2 in the S1 segment of proximal convoluted tubule (PCT) of nephrons, and subsequently transported to the basolateral interstitial fluid through GLUT2. In diabetic cardiomyopathy therapy, SGLT2is act in a canonical manner by preventing renal glucose reabsorption to reduce overall hyperglycemia but studies indicate a non-canonical mechanism of SGLT2is in the heart. Owing to the absence of SGLT2 channels in the heart, selective SGLT2is can act on prolonged activated NHE to reduce intracellular sodium ions and subsequently promote an alkaline intracellular pH, which has been reported to activate AMPK and inhibit mTORC1. SGLT2 inhibitors have also been reported to mimic an energy-deprived environment that might promote Sestrin2-mediated AMPK and SIRT1 activation, leading to mTORC1 inactivation and cardiovascular benefits. PCT—proximal convoluted tubule; GLUT2—glucose transporter 2; NHE—sodium–hydrogen exchanger; NCX—sodium–calcium exchanger. ↑Na_i_^+^/↑Ca_i_^2+^/↑mTORC1/↑ROS indicate an increase in intracellular Na^+^, Ca^2+^, mTORC1, and ROS levels; faded tail → through the NHE/GLUT2/SGLT2 channels indicates the flow of molecules/ions; dotted – → indicates indirect effects and solid → indicates direct activation/induction; blunt 
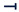
 indicates inhibition.

**Table 1 ijms-24-15078-t001:** Therapeutic targets of diabetic cardiomyopathy and associated clinical trials.

Therapeutic Target	Drug Treatment	Study	Design of Study	Study Outcomes
PPARα	Blinded fenofibrate or placebo plus simvastatin	ACCORD (1999–2012) [24,25]	Randomized, double-blind, placebo-controlled phase III trial in T2D patients (actual enrollment—10251)	For the primary outcome, cardiovascular risk was lower in the intense glycemia and blood pressure (BP) groups, compared to combined standard BP and glycemia treatment. For secondary outcomes, myocardial infarction and stroke were significantly reduced by intensive glycemia and BP treatment. There were more adverse effects but no statistically significant benefit or harm in terms of total mortality and cardiovascular disease mortality for any intensively treated groups compared to standard.
PDE5A	Sildenafil or placebo	CECSID (2008–2009) [26,27]	Randomized, double-blind, placebo-controlled phase IV trial in male T2D patients (Actual Enrollment- 59)	The study showed an improved ratio of left ventricular mass to end diastolic volume and LV contraction, besides reducing TGFβ levels and demonstrating an anti-remodeling effect. Endothelial function or cardiac metabolism were not affected, and no significant differences were found in glycemia, insulin, c-peptide, or lipid profile.
GLP1R	Liraglutide or placebo	LEADER (2010–2015) [28,29]	Multi-center, randomized, double-blind, placebo-controlled phase III trial in T2D patients (Actual Enrollment- 9341)	Liraglutide significantly reduced cardiovascular (CV) outcomes in patients with myocardial infarction (MI)/stroke history or having atherosclerotic CV diseases without MI/stroke history, but no improvement was reported in patients with only CV risk. In all the three groups, the percentage of adverse gastrointestinal events ranged from 55–65%.
IL-1β	Canakinumab or placebo or standard of care	CANTOS (2011–2019) [30,31]	Randomized, double-blind, placebo-controlled, event-driven phase III trial in patients with myocardial infarction and elevated hsCRP levels with/without T2D (actual enrollment—10066)	Canakinumab reduced hsCRP and IL6 levels in patients with or without diabetes, thereby reducing recurrent cardiovascular events and heart failure hospitalizations, but did not reduce new-onset diabetes. Furthermore, the treatment had no long-term benefits on HbA_1c_ or fasting plasma glucose.
NRF2	Sulforaphane or placebo	Clinical trial with broccoli sprout extract to patients with type 2 diabetes (2015–2020) [32,33]	Randomized, double-blind, placebo-controlled phase II trial in T2D patients (actual enrollment—103)	Sulforaphane improved HbA1c and fasting glucose levels in patients with obesity and T2D but the study was not focused on cardiovascular health or outcomes. No severe adverse effects were observed.

**Table 2 ijms-24-15078-t002:** Phase III clinical trials of SGLT2is for diabetic cardiac dysfunction.

Study and Duration	Treatment	Total Enrollment	Key Inclusion Criteria	Study Outcomes	Adverse/Side Effects	Study Limitations
EMPA-REG OUTCOME 2010–2015 [100,105,109,110]	Empagliflozin vs. placebo	7064	Patients with T2D and high-risk/established cardiovascular disorders.	Reduction in cardiovascular death and non-fatal myocardial infarction. For EMPA-REG post hoc analysis, refer to [111,112].	Moderate benign mycotic genital infections.	The study lacked adjustment for background medications in-trial and had a controversial post hoc nature [113,114].
CANVAS 2009–2017 [30,115]	Canagliflozin vs. placebo	4330	Patients with T2D and high cardiovascular risk; enrolled women population post-menopausal or on a birth-control regime.	Reduction in the composite of cardiovascular deaths, non-fatal myocardial infarction, and non-fatal stroke. For CANVAS post hoc analysis, refer to [116,117].	Risk of amputation (metatarsal) and moderate risk of genital infections.	The program had a relatively small participant proportion, indicating moderate number of events for health outcomes and increasing the risk of false positive findings [115].
DECLARE-TIMI 58 2013–2018 [118,119]	Dapagliflozin vs. placebo	17,190	Patients with diabetes mellitus and non-insulin-dependent cardiovascular risk.	Lower glycated hemoglobin along with lower rates of cardiovascular diseases and hospitalization. For DECLARE-TIMI post hoc analysis, refer to [120].	Moderate genital infections.	Low African American and Hispanic study population precludes any definite understanding of ethnicity-based treatment outcomes. Moreover, blood pressure subanalysis categories were not prespecified in the study [121,122].
VETRIS CV 2013–2019 [123,124]	Ertugliflozin vs. placebo with background glycemic rescue	8246	Patients with T2D and established cardiovascular diseases.	Incidence of cardiovascular deaths and heart failure hospitalizations did not differ significantly between the ertugliflozin and placebo groups. For VETRIS CV post hoc analysis, refer to [125].	Amputation risk in ~2% patients of the ertugliflozin groups.	The study population was predominantly white and male, limiting ethnicity and sex-based study interpretation. Moreover, differences in baseline characteristics in some subgroups might affect the influence of background medications on observed Ertugliflozin effects [126].
DAPA-HF 2017–2019 [106,127]	Dapagliflozin vs. placebo	4744	Patients with <40% ejection fraction and symptomatic heart failure; 50% of patients with T2D.	Reduction in cardiovascular deaths and heart failure hospitalizations for both diabetic and non-diabetic patients. For DAPA-HF post hoc analysis, refer to [128,129].	No significant excess of genital infection or amputations observed between the dapagliflozin and placebo groups.	The main limitation of the DAPA-HF trial included a reduced population of Black patients (<5%), elderly patients with co-morbidities (>66 years), and patients with sacubitril-valsartan at baseline [106].
EMPEROR-Reduced 2017–2020 [108,130]	Empagliflozin vs. placebo	3730	Patients with <40% ejection fraction and chronic heart failure risk; 50% of the patients with T2D.	Reduction in cardiovascular risk and heart failure hospitalizations in both diabetic and non-diabetic patients. For EMPEROR-Reduced post hoc analysis, refer to [131,132].	Uncomplicated genital tract infections observed in the empagliflozin group.	The median follow-up duration was very limited (16 months) and outpatient events were not adjudicated or reviewed [133].

## Data Availability

Not applicable.

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
