# Peer review of "mTORC1 and SGLT2 Inhibitors—A Therapeutic Perspective for Diabetic Cardiomyopathy"

_ijms, 2023, doi:10.3390/ijms242015078_

Round 1
Reviewer 1 Report
The area of selected by the authors is very timely, interesting and valuable. In this interesting and important review article, the authors present the cardiovascular aspects related to cardiac dysfunction and heart failure in diabetes in a fairly comprehensive manner. The figures and the section as proposed appear to be in line and consistent with the proposed topic. The article is very well written. Only minor comments listed below:
Comments:
(a) Around 18 million more men than women suffer from diabetes millitus. It would be appropriate to underline a characteristic aspect of the response to diabetes, linked to sex.
(b) It will be great if authors can comment on occurrence of type 2 diabetes in middle-aged and older people.
Author Response
Thank you for your valuable and insightful comments to improve the manuscript. We have added statistics regarding sex-differences in diabetes and the prevalence of diabetes in elderly people on Page1, Lines 35-37 of the revised manuscript. We hope that you will find our revised version satisfactory for publication in IJMS.
Best regards,
Anindita

Reviewer 2 Report
I'd like to commend you on your comprehensive review of the role of SGLT2 inhibitors in diabetic cardiomyopathy. Your work provides valuable insights into this emerging area of research.
Here are my detailed comments:
- Scope and Depth: The article provides a comprehensive overview of the role of SGLT2 inhibitors in diabetic cardiomyopathy. However, it would benefit from a deeper dive into the molecular mechanisms underlying the observed effects. While the article touches upon mTORC1 regulation and ionic dyshomeostasis, a more detailed exploration of these pathways would enhance the article's depth.
- SGLT2 Expression in the Heart: The article rightly points out that the expression of SGLT2 channels in the heart is negligible. This is a crucial point, as it underscores the non-canonical effects of SGLT2 inhibitors. However, a more detailed discussion on the potential off-target effects of these drugs, especially in the context of cardiac tissue, would be beneficial.
- Role of SGLT1: The mention of SGLT1's potential involvement in mediating the effects of certain SGLT2 inhibitors is intriguing. However, the article could delve deeper into the differential roles of SGLT1 and SGLT2 in both renal and cardiac tissues, providing a clearer picture of their distinct and overlapping functions.
- Mechanisms of Action: The article discusses several mechanisms through which SGLT2i might exert their cardioprotective effects, including ionic homeostasis and nutrient-deprivation mimicry. While these are valid hypotheses, the article would benefit from a more critical evaluation of the existing evidence supporting each mechanism, highlighting any contradictory findings in the literature.
- Clinical Trials: The table summarizing phase 3 clinical trials of SGLT2i for diabetic cardiac dysfunction is informative. However, a discussion on the limitations of these trials, potential biases, and any post-hoc analyses would provide a more balanced view.
- Adverse Effects: While the article mentions some adverse effects associated with SGLT2i, a more comprehensive discussion on the safety profile of these drugs, including rare but severe side effects, would be essential for a holistic understanding.
- Future Directions: The conclusion rightly emphasizes the need for further research to understand the non-glycemic effects of SGLT2i and their role in mTORC1/mTORC2 regulation. However, the article could benefit from a dedicated section discussing potential future research directions, emerging therapeutic strategies, and any ongoing clinical trials that readers should be aware of.
- Comparative Analysis: It would be beneficial to include a comparative analysis of SGLT2i with other anti-diabetic drugs, highlighting their unique advantages, potential synergistic effects, and any concerns regarding combination therapies.
- Concluding Remarks: The article concludes with a call for further research into the role of SGLT2i in diabetic cardiomyopathy. While this is a valid point, the conclusion could be strengthened by highlighting the potential clinical implications of these findings and their significance for patient care.
Author Response
We are very thankful to the reviewer for the valuable suggestions to improve our manuscript. We have revised the manuscript according to all suggested comments. We have addressed and highlighted the edited sections in the revised manuscript for your reference. The revisions are noted as follows:
- Scope and Depth: A more detailed context of the regulation of ionic homeostasis and mTOR signaling/SGLT2i has been included on Pages 7-8, Lines179-201 and Page 14, Lines 348-371.
- SGLT2 Expression in the Heart: information on the potential off-target effects of SGLT2i in the context of cardiac tissue including SGLT1, NHE, and possible GLUT-binding, have been included on Pages13-14, Lines 312-371 in the revised manuscript.
- Role of SGLT1: A brief introduction and roles of SGLT1 and SGLT2 in Kidney have been included on Pages 9-10, Lines 260-271, whereas the role of SGLT1 in heart has been mentioned on Page13, Lines 315-319 in the revised manuscript.
- Mechanisms of Action: The following revisions based on mechanism of SGLT2 action have been included in the revised manuscript: Context of ionic homeostasis-Page14 Lines 341-371; Context of nutrient deprivation- Pages14-15, Lines 373-398; Context of existing evidence of SGLT2i direct cardiac effects Pages16-17 Lines 414-448.
- Clinical Trials: The Table 2 has been revised with details on limitations and post-hoc analysis for each SGLT2i clinical trial.
- Adverse Effects: Aside from the mention of adverse effects of each SGLT2i clinical trial in Table 2, we have described the adverse effects on Page 10, Lines 297-309.
- Future Directions: A separate section of Future Directions has been included in the revised manuscript on Pages17-18, Lines 451-510.
- Comparative Analysis: A detailed comparative analysis between SGLT2i and other anti-diabetic drugs has been included on Page17, Lines 451-480 in the revised manuscript.
- Concluding Remarks: We have highlighted certain clinical implications and patient care concerns in the Conclusion section on Pages18-19, Lines 523-532.
We hope that you will find our revised version satisfactory for publication in IJMS.
Best regards,
Anindita
